# Learning Riemannian Stable Dynamical Systems via Diffeomorphisms

**Jiechao Zhang**[1, 2]     **Hadi Beik-Mohammadi**[1, 2]     **Leonel Rozo**[1]

[1]Bosch Center for Artificial Intelligence. Renningen, Germany.
[2.] Karlsruhe Institute of Technology (KIT), Karlsruhe, Germany.
leonel.rozo@de.bosch.com     hadi.beik-mohammadi@de.bosch.com

**Abstract:** Dexterous and autonomous robots should be capable of executing elaborated dynamical motions skillfully. Learning techniques may be leveraged to build models of such dynamic skills. To accomplish this, the learning model needs to encode a stable vector field that resembles the desired motion dynamics. This is challenging as the robot state does not evolve on a Euclidean space, and therefore the stability guarantees and vector field encoding need to account for the geometry arising from, for example, the orientation representation. To tackle this problem, we propose learning Riemannian stable dynamical systems (RSDS) from demonstrations, allowing us to account for different geometric constraints resulting from the dynamical system state representation. Our approach provides Lyapunov-stability guarantees on Riemannian manifolds that are enforced on the desired motion dynamics via diffeomorphisms built on neural manifold ODEs. We show that our Riemannian approach makes it possible to learn stable dynamical systems displaying complicated vector fields on both illustrative examples and real-world manipulation tasks, where Euclidean approximations fail.

**Keywords:** Dynamical systems, Riemannian manifolds, Motion learning

## 1   Introduction

The promise of having fully-autonomous robots performing a large variety of tasks implies that robots should be able to execute highly-dynamic motions. The inherent complexity of these movements makes hand coding an infeasible approach. Therefore learning techniques arise as a potential solution. In particular, learning dynamic robot motions from human demonstrations is a promising approach to build models of dynamic skills in an intuitive, sample-efficient and quick manner. However, learning dynamical motions is not trivial as the learning model requires to provide stability guarantees, which is also an intrinsic property in human motion generation [1]. In this context, most research works have focused on learning time-invariant stable dynamical systems for goal-driven motions (a.k.a point-to-point movements) with Lyapunov-stability guarantees [2, 3, 4, 5, 6].

Khansari-Zadeh and Billard [2] proposed one of the first approaches to learn stable dynamical systems from human demonstrations by imposing quadratic Lyapunov non-linear constraints on the model parameters' optimization, which limited the range of possible learnable motions. As the class of stable dynamical systems constrained by a predefined Lyapunov function is a subset of all possible stable dynamical systems, this limits the learned model accuracy [4]. To improve accuracy, a more general parametric control Lyapunov function [7] can be learned from demonstrations [3, 8, 9]. The trade-off between stability and accuracy motivated the use of diffeomorphisms [4, 5, 6, 10, 11, 12], which leveraged a more general class of stable dynamical systems. Their main idea is to design or learn a canonical Lyapunov-stable dynamical system on a latent space and use a diffeomorphic mapping to transform the demonstrations to the latent space so that they are consistent with the desired Lyapunov-stable behavior. Thus, the modeling accuracy depends on the expressiveness of the diffeormorphic function, often modeled by a neural network [5, 6, 10, 11].

Most of aforementioned works assume that the training data lie in the Euclidean space [4, 5, 6, 10, 12], with the exception of Urain et al. [11], which severely limits their use in real applications. For

6th Conference on Robot Learning (CoRL 2022), Auckland, New Zealand.

instance, there are various types of representation for the robot end-effector's orientation, namely unit quaternions in the 3-Sphere manifold [13], and rotation matrices in the special orthogonal group (SO(3)) manifold [14], which do not lie in the Euclidean space. Accounting for the data geometry has proven critical when learning and optimizing movement primitives on quaternion space [15, 16, 17, 18, 19], as relying on Euclidean approximations leads to modeling distortions and compromises extrapolation. The importance of geometry-aware methods when learning dynamical systems was recently addressed in [11], where a stable dynamical system was learned via diffeomorphisms over Lie groups. Although Lie theory has been exploited to operate with data of specific geometries [20], a potential limitation is that not all types of manifolds arising in robotics can be easily endowed with a Lie group structure (e.g., the space of symmetric positive-definite matrices (SPD)).

A more general solution based on Riemannian geometry [21] is proposed in this paper. We consider dynamical systems evolving on a Riemannian manifold. This arises two main challenges: *(1)* designing a canonical Lyapunov-stable dynamical system on Riemannian manifolds, and *(2)* learning a diffeomorphism that accounts for the Riemannian geometry. To address these challenges, we leverage the Lyapunov stability analysis on Riemannian manifolds [22, 23]. Moreover, we exploit neural ordinary differential equations (ODEs) on Riemannian manifolds [24, 25] for constructing the diffeomorphism to learn a Riemannnian stable dynamical system (RSDS). Unlike previous works using diffeomorphisms [4, 5, 6], our approach extends this concept to systems evolving on Riemannian manifolds. In contrast to works assuming Riemannian manifolds that are diffeomorphic to the Euclidean space [26], or manifold-specific diffeomorphisms built on specialized neural networks [27], our approach leverages a general formulation to construct diffeomorphisms based on solutions of ODEs evolving on arbitrary Riemannian manifolds [24]. Our approach is conceptually similar to the Lie-group method introduced in [11], as both explicitly consider the data geometry to design technically-sound learning models via diffeomorphisms. However, our Riemannian formulation substantially differs from [11] in its technical development, and provides a more general approach that may be exploited for a variety of Riemannian manifolds.

In summary, we propose a method to learn Riemannian stable dynamical systems from demonstrations. Our approach provides Lyapunov-stability guarantees on Riemannian manifolds (see § 3.1) that are enforced on the desired motion dynamics via diffeomorphisms built on neural manifold ODEs (see § 3.2 and 3.3). Through a set of evaluations on the 2-sphere manifold $\mathcal{S}^2$, presented in § 4, we show that our Riemannian approach is able to learn complicated dynamical systems, in contrast to Euclidean approximations which fail to encode stable vector fields. Also, we learn realistic motion skills on a 7-DoF robotic manipulator featuring complex full-pose trajectories on $\mathbb{R}^3 \times \mathcal{S}^3$.

## 2   Background

### 2.1   Dynamical Systems and Lyapunov Stability

We here give a short review of Lyapunov stability in the Euclidean setting. Let us assume an autonomous dynamical system $\dot{x} = f(x)$, with a single equilibrium point $x^*$, where $x \in \mathbb{R}^n$ is the state variable. Consider a potential function $V(x(t))$ describing the energy of such a system. If this system loses energy over time and the energy is never restored, the system must eventually reach some final resting state. This idea is formally described as (see [28] for details):

**Theorem 1** (Lyapunov Stability). *A dynamical system $\dot{x} = f(x)$ is globally asymptotically stable at $x^*$ if there exists a continuously differentiable Lyapunov function $V(x) : \mathbb{R}^n \to \mathbb{R}$ such that*

$$V(x^*) = 0, \quad \dot{V}(x^*) = 0, \quad V(x) > 0, \ \forall \, x \neq x^*, \quad \dot{V}(x) < 0, \ \forall \, x \neq x^*. \tag{1}$$

From Theorem 1 we know that we can always find a Lyapunov function that fulfills these four conditions in Eq. 1 for a globally asymptotically stable dynamical system.

### 2.2   Riemannian Manifolds

A smooth manifold $\mathcal{M}$ can be seen as a set of points that locally, but not globally, resemble the Euclidean space $\mathbb{R}^d$ [21, 29]. An abstract definition of a manifold specifies the topological, differential and geometric structure by using *charts*, which are maps between parts of $\mathcal{M}$ to $\mathbb{R}^d$. The collection

of these charts (a.k.a. local parameterizations) is called *atlas*. More formally, a chart on a smooth manifold $\mathcal{M}$ is a diffeomorphic mapping (i.e. a bijective and differentiable function) $\varphi : U \to \tilde{U}$ from an open set $U \subset \mathcal{M}$ to an open set $\tilde{U} \subseteq \mathbb{R}^d$ (see Fig. 6 in App. A.1). Moreover, the transition map between two intersecting sets $U_1$ and $U_2$, given by $\varphi_1 \circ \varphi_2^{-1}$ or $\varphi_2 \circ \varphi_1^{-1} : \mathbb{R}^d \to \mathbb{R}^d$ is also a diffeomorphism. The smooth structure of $\mathcal{M}$ makes it possible to take derivatives of curves on the manifold, leading to tangent vectors in $\mathbb{R}^d$. The set of tangent vectors of all curves at $\boldsymbol{x} \in \mathcal{M}$ spans a $d$-dimensional affine subspace of $\mathbb{R}^d$, which is known as the *tangent space* $\mathcal{T}_{\boldsymbol{x}}\mathcal{M}$ of $\mathcal{M}$ at $\boldsymbol{x}$. The collection of all tangent spaces of $\mathcal{M}$ is the *tangent bundle* $\mathcal{TM} = \bigsqcup_{\boldsymbol{x} \in \mathcal{M}} \mathcal{T}_x\mathcal{M}$. Therefore, a velocity vector $\dot{\boldsymbol{x}}$ at $\boldsymbol{x} \in \mathcal{M}$ lies on $\mathcal{T}_x\mathcal{M}$, and consequently a vector field on $\mathcal{M}$ lies on $\mathcal{TM}$.

The above definitions do not provide the mechanisms to measure how curved $\mathcal{M}$ is, or to compute distances on $\mathcal{M}$. To do so, we can endow $\mathcal{M}$ with a *Riemannian metric*, which is a family of inner products $g_{\boldsymbol{x}} : \mathcal{T}_{\boldsymbol{x}}\mathcal{M} \times \mathcal{T}_{\boldsymbol{x}}\mathcal{M} \to \mathbb{R}$ associated to each point $\boldsymbol{x} \in \mathcal{M}$. As a result, a *Riemannian manifold* $(\mathcal{M}, g)$ is a smooth manifold endowed with a Riemannian metric [29]. To operate with Riemannian manifolds, it is common practice to exploit the Euclidean tangent spaces. To do so, we resort to mappings back and forth between $\mathcal{T}_{\boldsymbol{x}}\mathcal{M}$ and $\mathcal{M}$ using the exponential and logarithmic maps. The exponential map $\mathrm{Exp}_{\boldsymbol{x}}(\boldsymbol{u}) : \mathcal{T}_{\boldsymbol{x}}\mathcal{M} \to \mathcal{M}$ maps a point $\boldsymbol{u}$ in the tangent space of $\boldsymbol{x}$ to a point $\boldsymbol{y}$ on the manifold, so that it lies on the geodesic starting at $\boldsymbol{x}$ in the direction $\boldsymbol{u}$, and such that the geodesic distance $d_{\mathcal{M}}(\boldsymbol{x}, \boldsymbol{y}) = d_{\mathbb{R}}(\boldsymbol{x}, \boldsymbol{u})$. The inverse operation is the logarithmic map $\mathrm{Log}_{\boldsymbol{x}}(\boldsymbol{y}) : \mathcal{M} \to \mathcal{T}_{\boldsymbol{x}}\mathcal{M}$. We provide all the necessary operations in App. A.1.

## 2.3 Diffeomorphism

A diffeomorphism $\psi : \mathcal{M} \to \mathcal{N}$ is a smooth bijective mapping between two smooth manifolds which preserves the topological properties of $\mathcal{M}$, and whose inverse $\psi^{-1}$ is also smooth. When learning stable dynamical systems, diffeomorphisms can be exploited to impose Lyapunov stability guarantees by transferring a manually-designed stable dynamical system on $\mathcal{N}$ to the desired manifold $\mathcal{M}$. We focus on constructing learnable diffeomorphisms that resemble continuous normalizing flows (CNFs) [30, 31, 32], which are bijective and bidirectionally differentiable mappings, and have been recently exploited on density estimation problems [33, 34, 35]. We here exploit them to learn diffeomorphic mappings between Riemannian manifolds.

Most CNFs are constructed from *neural ordinary differential equation* (Neural ODEs) in Euclidean space [30, 31, 32], with the exception of *neural manifold ordinary differential equations* (Neural MODEs) on Riemannian manifolds [24, 25]. Generally, Neural ODEs parametrize the dynamics of a hidden variable using a continuous-time ODE represented by a neural network, as follows,

$$\dot{\boldsymbol{z}}(t) = f_{\boldsymbol{\theta}}(\boldsymbol{z}(t), t), \tag{2}$$

where $\boldsymbol{z} \in \mathbb{R}^d$ is the state variable and $f_{\boldsymbol{\theta}} : \mathbb{R}^d \times \mathbb{R} \to \mathbb{R}^d$ is a neural network. According to Mathieu and Nickel [25] (see Theorem 2 in App. A.2), we can extend CNFs to the Riemannian setting, where the state variable $\boldsymbol{z} \in \mathcal{M}$ and the vector field $f_{\boldsymbol{\theta}} : \mathcal{M} \times \mathbb{R} \to \mathcal{TM}$. As a result, we can use (2) as a manifold ODE, whose initial value problem (IVP) solution results in a diffeormorphic mapping $\psi_{\boldsymbol{\theta}} : \mathcal{M} \to \mathcal{N}$, $\boldsymbol{x} = \boldsymbol{z}(t_s) \in \mathcal{M}$ and $\boldsymbol{y} = \boldsymbol{z}(t_e) \in \mathcal{N}$. i.e. $\boldsymbol{y} = \psi_{\boldsymbol{\theta}}(\boldsymbol{x}) = \boldsymbol{x} + \int_{\tau=t_s}^{t_e} f_{\boldsymbol{\theta}}(\boldsymbol{z}(\tau), \tau)d\tau$. To solve the IVP on $\mathcal{M}$, we leverage integrators on Riemannian manifolds based on the local representation via coordinate charts [36]. This method uses a local representation of $\mathcal{M}$ defined by a coordinate map $\varphi : \mathcal{M} \supseteq U \to \tilde{U} \subseteq \mathbb{R}^d$ with coordinates $\boldsymbol{w}(t) = \varphi(\boldsymbol{z}(t))$. Computing integrators on $\mathcal{M}$ can be approximated by solving an equivalent ODE in $\mathbb{R}^d$

$$\dot{\boldsymbol{w}}(t) = D_{\varphi^{-1}(\boldsymbol{w}(t))}\varphi \circ f_{\boldsymbol{\theta}}(\varphi^{-1}(\boldsymbol{w}(t)), t), \tag{3}$$

where $D_{\varphi^{-1}(\boldsymbol{w}(t))}\varphi$ represents the differential of $\varphi$ at $\varphi^{-1}(\boldsymbol{w}(t))$ (see App. A.2 and A.3 for details). Additionally, we use the adjoint method [30] on Riemannian manifolds [24] to compute gradients, which can also be used for calculating differentials. Consider a loss function $\mathcal{L} : \mathcal{M} \to \mathbb{R}$, in order to compute the derivative of $\mathcal{L}$ with respect to an intermediate variable $\boldsymbol{z}(t)$ of the manifold ODE, we can solve $\dot{\boldsymbol{a}}(t)^{\mathsf{T}} = -\boldsymbol{a}(t)^{\mathsf{T}} D_{\boldsymbol{z}(t)} f_{\boldsymbol{\theta}}(\boldsymbol{z}(t), t)$, where $\boldsymbol{a}(t)^{\mathsf{T}} := D_{\boldsymbol{z}(t)}\mathcal{L}$ (as detailed in App. A.4).

# 3 Learning Riemannian Stable Dynamical Systems

We here introduce our approach for learning stable dynamical systems on Riemannian manifolds from demonstrated point-to-point motions. First, let us consider that the recorded demonstrations follow a dynamical system $\dot{x} = f(x)$, where the state $x$ evolves on a Riemannian manifold $\mathcal{M}$ with velocity $\dot{x} \in \mathcal{T}_x\mathcal{M}$. This dynamical system is equivalent, under a change of coordinates, to another system defined on a latent Riemannian manifold $\mathcal{N}$. Under the diffeomorphism $\psi_{\theta} : x \mapsto y \in \mathcal{N}$, parameterized by $\theta$, we map the observed states $x$ onto $\mathcal{N}$. Then, we evaluate the canonical stable vector field $g_{\gamma}(y)$ to obtain the velocity $\dot{y} \in \mathcal{T}_y\mathcal{N}$. Finally, we

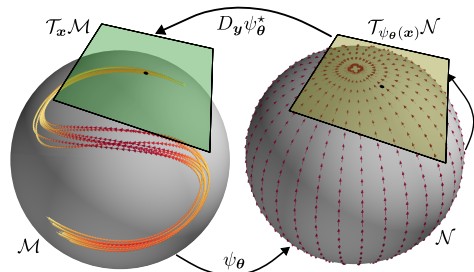

Figure 1: Architecture of a diffeomorphism-based stable vector field on the Riemannian manifold $\mathcal{S}^2$.

leverage the *pullback operator* $D_y\psi_{\theta}^{\star}$ to project $\dot{y}$ back to the tangent space $\mathcal{T}_x\mathcal{M}$. The whole procedure can be expressed as follows,

$$\dot{x} = (D_y\psi_{\theta}^{\star} \circ g_{\gamma} \circ \psi_{\theta})(x) = D_y\psi_{\theta}^{\star}(\dot{y}), \tag{4}$$

which is illustrated in Fig. 1 (a proof is given in App. B). In the sequel, we describe how we design a Lyapunov-stable vector field $g_{\gamma}$ on $\mathcal{N}$ to provide stability guarantees on the learned dynamical system. Later, we explain how to compute the diffeomorphism between the target and latent manifolds. Finally, we introduce two different methods to compute the pullback operator $D_y\psi_{\theta}^{\star}$.

## 3.1 Lyapunov-stable Geodesic Vector Fields

To design a stable vector field on the Riemannian manifold $\mathcal{N}$, we enforce the canonical dynamical system to follow geodesic curves that converge to a single equilibrium. Such a vector field can be designed via the logarithmic map. Specifically, given an equilibrium point $y^* \in \mathcal{N}$, the corresponding velocity vector $\dot{y} \in \mathcal{T}_y\mathcal{N}$ can be computed as $\dot{y} = g_{\gamma}(y) = k_{\gamma}(y)g_n(y)$ with the normalized geodesics vector field $g_n(y) := \frac{\mathrm{Log}_y(y^*)}{\|\mathrm{Log}_y(y^*)\|_2}$. This implies that the direction of tangent vectors is fully specified by $\mathrm{Log}_y(y^*)$, while their magnitude depends on the scaling factor $k_{\gamma} : \mathbb{R}^n \supset \mathcal{N} \to \mathbb{R}_{\geq 0}$. We can prove the stability of this geodesic vector field by choosing the Lyapunov function $V(y) := \langle F, F \rangle_{y^*}$ with $F = \mathrm{Log}_{y^*}(y)$, and applying Theorem 3 for Lyapunov stability on Riemannian manifolds, as detailed in App. B. Given that our geodesic vector field is Lyapunov stable, we can easily prove that the desired dynamical system is also globally asymptotically stable by defining a new valid Lyapunov function $\tilde{V}(x) := V(\psi_{\theta}(x))$ via the diffeomorphism $\psi_{\theta}$, with a single equilibrium point $x^* = \psi_{\theta}^{-1}(y^*) \in \mathcal{M}$. As $\psi_{\theta}$ preserves the topological properties of $\mathcal{N}$, the equilibrium point $x^*$ is also globally asymptotically stable on $\mathcal{M}$ (see App. B for the proof). Note that for certain Riemannian manifolds, it is only possible to guarantee *quasi-global* stability guarantees due to the Poincaré-Hopf theorem (see App. B for details).

Note that we separate the parameterization for the magnitude and direction of vector fields to improve the expressiveness of our framework. By relocating the scaling factor $k_{\gamma}$ and normalizing the vector fields governed by (4), we can obtain our final RSDS learning framework

$$\dot{x} = \hat{k}_{\gamma}(x) \frac{(D_y\psi_{\theta}^{\star} \circ g_n \circ \psi_{\theta})(x)}{\|(D_y\psi_{\theta}^{\star} \circ g_n \circ \psi_{\theta})(x)\|_2}, \tag{5}$$

where $\hat{k}_{\gamma}(x)$ is the new positive scaling factor that fully determines the magnitude of the learned vector fields. In App. C, we prove that the models (4) and (5) are equivalent.

## 3.2 Diffeomorphisms on Riemannian Manifolds

Given the final RSDS in (5) and $M$ demonstrations, the goal of learning stable dynamics on a Riemannian manifold reduces to learning $\psi_{\theta}$, computing its pullback operator $D_y\psi_{\theta}^{\star}$, and subsequently estimating $\hat{k}_{\gamma}(x)$. However, due to the geometric constraints arising from $\mathcal{M}$, learning a diffeomorphism and calculating the corresponding pullback operator are non-trivial problems. To address them, we leverage Neural MODEs [24] to build the diffeomorphism $\psi_{\theta}$. Unlike [24], we propose a

novel approach to compute the pullback operator by reversing the time interval of the ODE integration (see § 3.3), avoiding to explicitly compute the corresponding inverse. We also propose a method to design Lyapunov-stable geodesic vector fields on a Riemannian manifold, which are leveraged to provide stability guarantees on the learned dynamical system, as explained in § 3.1.

According to Theorem 2 in App. A.2, the dynamics $f_{\boldsymbol{\theta}}$ of Neural MODEs only has to be a $\mathcal{C}^1$ function. To compute the diffeomorphism with a parametric Neural MODE, we solve an integration problem based on the local parameterization $\boldsymbol{w}(t) = \varphi(\boldsymbol{z}(t))$ (described in App. A.3). Using this method requires the selection of coordinate charts, which can be created via the exponential map $\varphi_i^{-1} = \operatorname{Exp}_{\boldsymbol{z}_i}$ and logarithmic map $\varphi_i = \operatorname{Log}_{\boldsymbol{z}_i}$, similarly to [24]. Under this choice of coordinate mapping and given a fixed number of charts $k$, the diffeomorphism $\psi_{\boldsymbol{\theta}} : \boldsymbol{x} = \boldsymbol{z}_0 \mapsto \boldsymbol{z}_k = \boldsymbol{y}$, obtained via integration on the manifold can be then viewed as the composition of blocks of solving Neural ODEs and chart transitions defined as,

$$\psi_{\boldsymbol{\theta}} = \operatorname{Exp}_{\boldsymbol{z}_{k-1}} \circ \hat{\psi}_{\boldsymbol{\theta},k-1} \circ \operatorname{Log}_{\boldsymbol{z}_{k-1}} \circ \ldots \circ \operatorname{Exp}_{\boldsymbol{z}_0} \circ \hat{\psi}_{\boldsymbol{\theta},0} \circ \operatorname{Log}_{\boldsymbol{z}_0}, \quad \text{with}$$

$$\hat{\psi}_{\boldsymbol{\theta},i}(\boldsymbol{w}_i(t_{i,s})) = \boldsymbol{w}_i(t_{i,s}) + \int_{\tau=t_{i,s}}^{t_{i,e}} D_{\varphi_i^{-1}(\boldsymbol{w}_i(\tau))}\varphi_i \circ f_{\boldsymbol{\theta}}(\varphi_i^{-1}(\boldsymbol{w}_i(\tau)), \tau)d\tau, \tag{6}$$

where $i$ is the chart index, $t_{i,s}$ and $t_{i,e}$ are the starting and end time for $i^{th}$ chart. $\hat{\psi}_{\boldsymbol{\theta},i}$ defines a diffeomorphism computed by the classical ODE solver on the tangent space (i.e. Euclidean space) and provides the solution of the IVP of the equivalent ODE (3).

## 3.3 Differential of the Inverse Diffeomorphism

We are now left with the problem of computing the pullback operator $D_{\boldsymbol{y}}\psi_{\boldsymbol{\theta}}^{\star}$ in (4), which maps the latent velocity $\dot{\boldsymbol{y}}$ back to the original tangent space $\mathcal{T}_{\boldsymbol{x}}\mathcal{M}$. This operator can be considered as the inverse mapping of the differential $D_{\boldsymbol{x}}\psi_{\boldsymbol{\theta}} : \mathcal{T}_{\boldsymbol{x}}\mathcal{M} \to \mathcal{T}_{\psi_{\boldsymbol{\theta}}(\boldsymbol{x})}\mathcal{N}$. As we already have the diffeomorphism $\psi_{\boldsymbol{\theta}}$, the straightforward solution is to compute its derivatives and then obtain the required inverse. Nevertheless, under the Riemannian setting, particularly for $d$-dimensional submanifolds $\mathcal{M}^d$ embedded in $\mathbb{R}^n$, computing the inverse directly becomes problematic due to the geometric constraints arising from $\mathcal{M}$. Next, we provide two methods to deal with this problem.

**Pullback operator via constrained optimization:** Instead of naively differentiating through the ODE solver of $\psi_{\boldsymbol{\theta}}$, we can use the adjoint method to calculate the differential of a diffeomorphism constructed by a Neural MODE. Assuming that we have the differential $D_{\boldsymbol{x}}\psi_{\boldsymbol{\theta}}$ (as computed in Algorithm 1 in App. A.4), the connection between tangent vectors $\dot{\boldsymbol{x}}$ and $\dot{\boldsymbol{y}}$ can be written as $D_{\boldsymbol{x}}\psi_{\boldsymbol{\theta}}(\boldsymbol{x})\dot{\boldsymbol{x}} = \dot{\boldsymbol{y}}$. In the Euclidean case, we can directly compute $\dot{\boldsymbol{x}} = (D_{\boldsymbol{x}}\psi_{\boldsymbol{\theta}}(\boldsymbol{x}))^{-1}\dot{\boldsymbol{y}}$. However, under the Riemannian setting, computing the inverse $(D_{\boldsymbol{x}}\psi_{\boldsymbol{\theta}}(\boldsymbol{x}))^{-1}$ often leads to a loss of rank in the matrix representation of $D_{\boldsymbol{x}}\psi_{\boldsymbol{\theta}}(\boldsymbol{x})$ for an embedded submanifold $\mathcal{M}^d$ due to the intrinsic geometric constraints of $\boldsymbol{x}$. We address this problem by introducing geometric constraints that allow us to compute $\dot{\boldsymbol{x}}$ on $\mathcal{T}_{\boldsymbol{x}}\mathcal{M}$. For example, for manifold $\mathcal{S}^d$, the tangent vector $\dot{\boldsymbol{x}}$ is orthogonal to $\boldsymbol{x}$, that is $\boldsymbol{x}^{\mathsf{T}}\dot{\boldsymbol{x}} = 0$. Hence, we can find a solution by solving a constrained optimization problem, from which the pullback operator $D_{\boldsymbol{y}}\psi_{\boldsymbol{\theta}}^{\star}$ is obtained as,

$$D_{\boldsymbol{y}}\psi_{\boldsymbol{\theta}}^{\star} = \left[ D_{\boldsymbol{x}}\psi_{\boldsymbol{\theta}}(\boldsymbol{x})^{\mathsf{T}} D_{\boldsymbol{x}}\psi_{\boldsymbol{\theta}}(\boldsymbol{x}) + \boldsymbol{x}\boldsymbol{x}^{\mathsf{T}} \right]^{-1} D_{\boldsymbol{x}}\psi_{\boldsymbol{\theta}}(\boldsymbol{x})^{\mathsf{T}}. \tag{7}$$

The full derivation and discussions can be found in App. D.1. Note that $D_{\boldsymbol{y}}\psi_{\boldsymbol{\theta}}^{\star}$ in (7) is specific to hypersphere manifolds due to the choice of constraints. Thus, this constrained optimization approach does not easily scale to compute the pullback operator for arbitrary Riemannian manifolds.

**Pullback operator via the adjoint method:** To generalize the computation of the pullback operator for arbitrary Riemannian manifolds, we introduce a new approach based on a modified version of the adjoint method. By reversing the integration time interval (i.e. from $[t_s, t_e]$ to $[t_e, t_s]$), we can determine the inverse diffeomorphism $\psi_{\boldsymbol{\theta}}^{-1}$, which is a distinct benefit of Neural MODEs. Thus, the pullback operator $D_{\boldsymbol{y}}\psi_{\boldsymbol{\theta}}^{\star}$ can be viewed as the differential of the inverse diffeomorphism $D_{\boldsymbol{y}}(\psi_{\boldsymbol{\theta}}^{-1})$. Furthermore, we leverage the adjoint method to compute the differential of $\psi_{\boldsymbol{\theta}}^{-1}$ using the adjoint ODE $\dot{\boldsymbol{A}}^*(t) = -\boldsymbol{A}^*(t)D_{\boldsymbol{z}(t)}f_{\boldsymbol{\theta}}(\boldsymbol{z}(t), t)$, with $\boldsymbol{A}^*(t) := D_{\boldsymbol{z}(t)}(\psi_{\boldsymbol{\theta}}^{-1})$. Due to the availability of starting states $\boldsymbol{z}(t_s) = \boldsymbol{x}$ and $\boldsymbol{A}^*(t_s) = \boldsymbol{I}_n$, we can integrate both the Neural MODE (2) and adjoint ODE to get the $D_{\boldsymbol{y}}(\psi_{\boldsymbol{\theta}}^{-1})$. For clarification, we provide Algorithm 2 in App. D.2 for computing

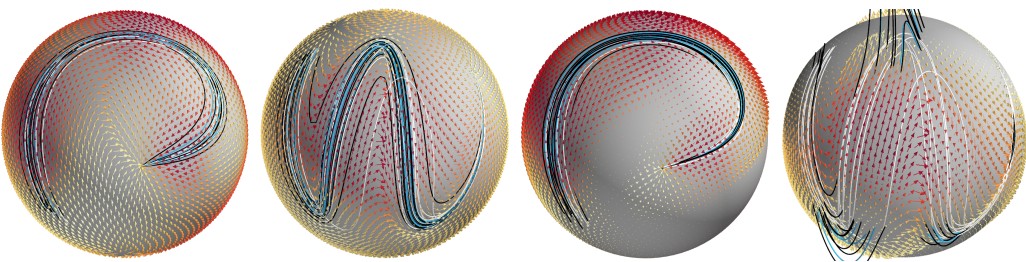

Figure 2: Experiments on LASA dataset on $\mathcal{S}^2$ (P and W letters): Demonstrations (white), learned vector fields, and reproductions (black and blue). Blue trajectories start at the same initial points as the demonstrations, while the black ones depart from randomly-sampled points around the initial points of the demonstrations. The first two plots show the results for our RSDS approach, and the last two display the *EuclideanFlow* results.

$D_{\boldsymbol{y}}(\psi_{\boldsymbol{\theta}}^{-1})$. Although, we can use dynamic charts method to solve the Neural MODE, dealing with the adjoint ODE dynamics is still not straightforward. The main challenge is to compute the differential of the vector fields on the Riemannian manifold $D_{\boldsymbol{z}(t)}f_{\boldsymbol{\theta}}(\boldsymbol{z}(t), t)$, despite it is nothing but partial derivatives in the Euclidean case. To avoid directly computing the differential of vector fields on $\mathcal{M}$, we adopt an approach similar to (6), such that a component $\boldsymbol{z}_i = (\mathrm{Exp}_{\boldsymbol{z}_i} \circ \hat{\psi}_{\boldsymbol{\theta},i}^{-1} \circ \mathrm{Log}_{\boldsymbol{z}_i})\boldsymbol{z}_{i+1}$ for computing the inverse diffeomorphism has its differential as,

$$D_{\boldsymbol{z}_{i+1}}\boldsymbol{z}_i = D_{\boldsymbol{w}_i(t_{i,s})}\mathrm{Exp}_{\boldsymbol{z}_i} \circ D_{\boldsymbol{w}_i(t_{i,e})}\hat{\psi}_{\boldsymbol{\theta},i}^{-1} \circ D_{\boldsymbol{z}_{i+1}}\mathrm{Log}_{\boldsymbol{z}_i}, \tag{8}$$

where $D_{\boldsymbol{w}_i(t_{i,e})}\hat{\psi}_{\boldsymbol{\theta},i}^{-1}$ boils down to partial derivatives (the proof of (8) is provided in App. D.2).

## 4 Experiments

We evaluate our method in two settings: reproducing trajectories based on the LASA dataset [29] projected on $\mathcal{S}^2$; and reproducing real dynamic motions learned from demonstrations. To show the importance of incorporating geometry, we compare against a baseline method similar to Euclideanizing flows [5], that is implemented using CNFs with Neural ODEs in $\mathbb{R}^n$ for illustrative experiments on a modified LASA dataset. We refer to this baseline as *EuclideanFlow* and our model as *RSDS*.

### 4.1 LASA dataset on $\mathcal{S}^2$

**Architectures:** For *EuclideanFlow*, we use a fully-connected neural network with an input vector on $\mathbb{R}^4$ (i.e., the 3-dimensional state $\boldsymbol{x}$ and time), and 3 hidden layers each, with 32 hidden units for $\mathcal{S}^2$. We use $\tanh$ as activation function to guarantee a $\mathcal{C}^1$-bounded mapping for modeling the Neural ODE. The RSDS architecture has an additional projection operator $\mathrm{proju}$ on the head of the network to impose the output on the tangent space of manifolds. The scaling factor $\hat{k}_{\boldsymbol{\gamma}}$ is generated using a network composed of an RBF layer and a linear layer without bias. The network architectures are depicted in App. E (Fig. 8). For all the experiments, we use an Euler ODE solver with step size of $1/32$, and 4 coordinate charts for the integration. For each dataset, one trajectory is used for testing and the remaining ones as training set. All models are trained using the ADAM optimizer [37] with learning rate of $10^{-3}$ (decaying by factor 0.1 after 1000 epochs) over 2000 epochs.

To illustratively show how RSDS and *EuclideanFlow* perform, we use a modified version of the LASA dataset of hand-written letters [38], whose trajectories are projected on $\mathcal{S}^2$. The corresponding vector field can be easily computed from the projected trajectories by using the logarithmic map. As a result, we have a new dataset $\{\{\boldsymbol{x}_{m,t}, \dot{\boldsymbol{x}}_{m,t}\}_{t=1}^{T_m}\}_{m=1}^M$ of $M$ demonstrations for each letter, with positions $\boldsymbol{x} \in \mathcal{S}^2$ and velocities $\dot{\boldsymbol{x}} \in \mathcal{T}_{\boldsymbol{x}}\mathcal{S}^2$, from which we learn stable dynamical systems. Figure 2 shows the resulting demonstrations as white curves for datasets P and W, the corresponding learned vector fields and the reproduced trajectories. Note that we provide more results in App. F and comparisons to an alternative method to *EuclideanFlow*, which projects trajectories onto the manifold after computing the integration in Euclidean space. Concerning the reproductions, the blue and black trajectories are rollouts starting from the same initial position as the demonstrations and from randomly-sampled points around them, respectively. Regarding our RSDS approach (first two plots in Fig. 2), it is evident that all blue and black rollouts closely match the demonstrations

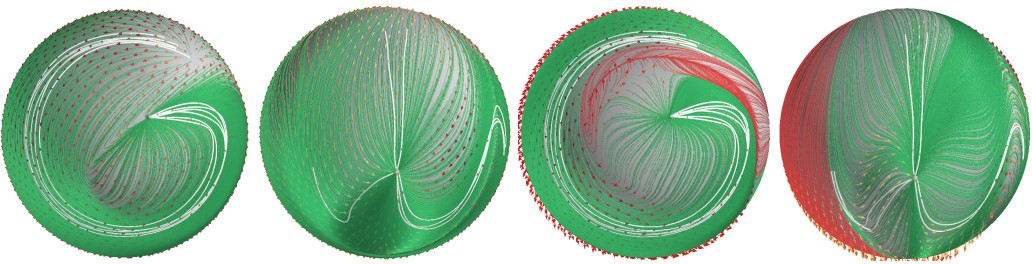

Figure 3: Stability of reproductions on $\mathcal{S}^2$ for G and MultiModels: 1000 trajectories starting from uniformly-sampled points. The successful and failed trajectories are depicted in green and red, respectively. The first two spheres from the left correspond to RSDS reproductions while the other two relate to *EuclideanFlow* results. For *EuclideanFlow*, we first project the vector fields onto $\mathcal{S}^2$ and then compute the integration trajectories.

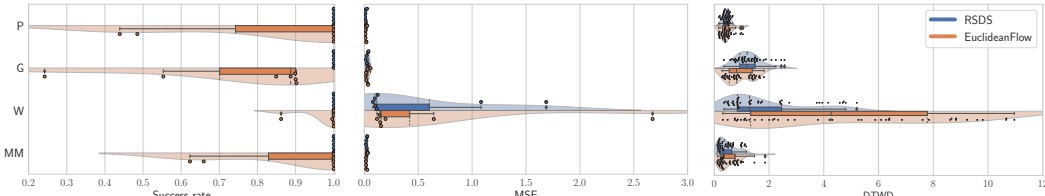

Figure 4: *Left*: Average success rate of RSDS and *EuclideanFlow* over randomly-sampled initial points on $\mathcal{S}^2$ using 7 different trained models indicated as points. *Middle:* Average mean square error (MSE) between observed and predicted velocity over data points in the test trajectories indicated as points. *Right:* Dynamic time warping distance (DTWD) between demonstrations and reproductions.

and converge to the equilibrium. In contrast, the *EuclideanFlow* reproductions constantly leave the manifold since there are no mechanisms accounting for the inherent geometric constraints of the data (see last two plots in Fig. 2). The bald regions on the manifold where the velocity vectors point inwards towards the sphere's center are also evidence of this phenomenon. We also provide quantitative metrics for accuracy comparisons. Figure 4-*right*, and -*middle* show the *dynamic time warping distance* (DTWD) as a measure of reproduced position trajectory accuracy, and the *mean squared error* (MSE) of the velocities reproduction. These metrics show that RSDS outperforms *EuclideanFlow* for the most complex trajectories, e.g. the W dataset. Although Fig. 4 shows that both models seem to perform well on the P, G, and MultiModels datasets, as pointed out before, Fig. 2 displays that the *EuclideanFlow* trajectories do not obey the data geometry.

Secondly, we evaluate the stability of both approaches. For a fair comparison, we first project the vector fields onto $\mathcal{S}^2$ and compute the integration trajectories for *EuclideanFlow*. To quantitatively assess this, we measure the stability of the learned vector fields (i.e. convergence to the equilibrium), by uniformly sampling 1000 initial points on $\mathcal{S}^2$ and counting the number of trajectories that successfully converge. This procedure is repeated for 7 different trained models, with the average success rate computed over the initial points. Using one of the trained models, Fig. 3 shows green and red curves representing successful and failed trajectories, respectively. It is evident that a large number of the *EuclideanFlows* trajectories failed to converge despite the projection, however all the *RSDS* trajectories succeeded. This result is supported by the success rate metric displayed in Fig. 4–*left*. These results show that accounting for the data geometry, as in our RSDS approach, is crucial to provide stability guarantees of the learned dynamical system. Additionally, we also compare the learning efficiency between the two methods, where RSDS generally requires fewer training epochs than *EuclideanFlow*. Nevertheless, each training epoch of *RSDS* is more computationally expensive due to manifold operations. The details of these results are provided in App. F (Fig. 10).

## 4.2 Real robot experiments on $\mathbb{R}^3 \times \mathcal{S}^3$

We evaluated two different manipulation tasks using the 7-DOF Franka-Emika Panda robot: *(1)* a GraspingTask with 90 degrees rotation, and *(2)* a V-shape DrawingTask. For both experiments, we collected 10 kinesthetic demonstrations of the robot end-effector motion as position-velocity trajectories at a frequency of 10 Hz. Here, we show that RSDS can indeed be used in real-world

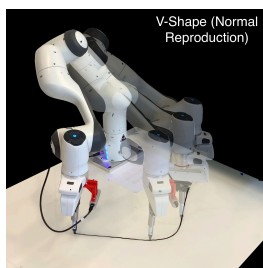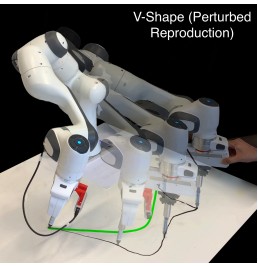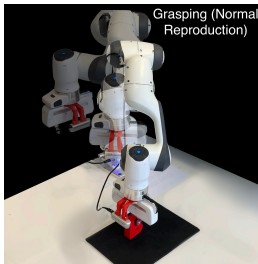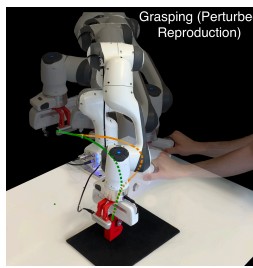

Figure 5: DrawingTask and GraspingTask: Time evolution of the reproductions is depicted by superimposed images from different time frames. The transparent robots depict the trace of the motion trajectory. The second plot for each task displays the task reproduction under perturbation, where the unperturbed reproduction is depicted as a green trajectory for reference.

applications. We train our model on the $\mathbb{R}^3 \times \mathcal{S}^3$ manifold accounting for position and orientation of robot end-effector with the same network architecture as our illustrative experiments, except that we use 16 hidden units for faster computations. As shown in Fig. 5, while following the V-shape curve in the DrawingTask, the end-effector always faces the moving direction; similarly, when approaching the object in the GraspingTask, the gripper rotates 90 degrees. These motion patterns require synchronized position and orientation trajectories, which is only attainable by training a model with a state variable on a product manifold, i.e., $\boldsymbol{x} \in \mathbb{R}^3 \times \mathcal{S}^3$. To deploy the reproduced motion on the robot, we numerically integrated the desired velocity vector $\hat{\boldsymbol{x}} \in \mathcal{T}_{\dot{\boldsymbol{x}}}\mathcal{M}$ online, and used it as reference for a Cartesian impedance controller.

As observed in Fig. 5, the reproductions governed by the vector fields learned with our RSDS model accurately imitate the demonstrated motion patterns and converge to the goal position. Furthermore, these experiments incorporated some target shifts to test if our model could cope with them without retraining. We further evaluated the stability of the learned vector fields by perturbing the robot during the task reproduction. As Fig. 5 shows, after perturbing the robot, the end-effector still follows an alternative trajectory computed from the learned vector field (see black and orange trajectories for DrawingTask and GraspingTask, respectively).

## 5 Discussion

We introduced a new approach RSDS to accurately learn vector fields on Riemannian manifolds while ensuring global asymptotic stability, which can not be achieved without taking into account the underlying geometry structure of the data. Our model inherits all the advantages of stable dynamical systems, such as high robustness against environmental perturbations. To our knowledge, RSDS is the first to leverage neural ODEs on Riemannian manifolds to learn Lyapunov-stable Riemannian dynamical systems. Moreover, RSDS builds on a new methodology to compute the pullback operator leveraging the characteristics of neural MODEs. Our framework is generic and can theoretically be used to learn vector fields on any Riemannian manifolds with defined exponential and logarithmic maps. As future work, we will leverage RSDS to learn vector fields on other Riemannian manifolds such as the manifold of symmetric-positive-definite (SPD) matrices $\mathcal{S}^d_{++}$, which is relevant in manipulability learning [23] and video tracking [39].

**Limitations:** Due to the complexity of the Riemannian operators and the Neural MODE solvers, our framework runs relatively slowly, making it unsuitable for hard real-time applications. This problem can be alleviated by switching to faster ODE solvers after training, which allows us to accelerate the query time at the expense of precision. To improve the accuracy and stability of solving Neural MODEs, we can take advantage of techniques such as regularization [32] and recording checkpoint for the forward mode [40]. In addition, since we leverage the Lyapunov stability to a single fixed point, the model may still reproduce some trajectories that are inconsistent with the trend of demonstration data due to lack of information for points far from demonstrations. It may be worthwhile exploring other stability criteria, such as contraction analysis [41], to ensure the incremental exponential stability of trajectories with respect to each other on the manifold.

**Acknowledgments**

J. Zhang was supported by the Bosch Center for Artificial Intelligence (BCAI) as a master thesis student.

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
