# OpenReview forum: "Learning Riemannian Stable Dynamical Systems via Diffeomorphisms"
_robot-learning.org/CoRL/2022/Conference — CoRL 2022 Poster_

### Official Review · Reviewer_AVAQ · 2022-07-27

**Originality:** Good
**Technical Quality:** Very Good
**Clarity Of Presentation:** Good
**Impact:** 3

**Recommendation:**

Weak Accept: I recommend accepting the paper, but will not argue for my recommendation if the majority of other reviewers have a different opinion.

**Summary:**

This work seeks to learn motion dynamics from human demonstrations. The work resembles Neumann et al., 2015, such that they view human demonstrations as trajectories from a stable dynamical system on a Riemannian manifold and enforce stability by mapping the dynamics into a latent space whose dynamics are manually-designed to be stable. As such, learning stable dynamics reduces to learning a diffeomorphism between the observed and latent manifolds. This paper extends the methods from Neumann et al., 2015, by encoding stable dynamics on a latent Riemannian manifold using geodesic vector fields, which they show to be Lyapunov stable. This forms one of their main contributions, along with a method for learning the diffeomorphism using Neural Manifold ODEs, which allows them to calculate the inverse diffeomorphism by simply reversing the integration time interval. They demonstrate their method on the LASA dataset projected onto the 2-sphere and on two real-world manipulator experiments.

**Issues:**

1. Incorrect statement about Euclideanizing Flows in Neumann et al., 2015 (citation 5 in paper) on line 39.
    * Neumann et al., 2015 do not assume the data lies in the Euclidean space.
    * On the contrary, they assume the data lies on a Riemannian manifold and use the diffeomorphism to map the system to a simpler Euclidean space where they can enforce stability.
2. RSDS should be compared to the Euclideanizing Flows with a Neural Manifold ODE to enable the importance of the Riemannian latent space to be verified.
3. I think the authors should be careful stating that they learn diffeomorphisms with normalising flows.
    * In my eyes, this method is inspired (or resembles) normalising flows. This is because it is not concerned with normalising density functions. Instead, it is interested in mapping complicated human motion trajectories (on an observed Riemannian manifold) to geodesic trajectories (on a latent Riemannian manifold) by sequentially applying the change of variable theorem.
    * Euclideanizing Flows received its name as it maps to a latent Euclidean space, just as Normalizing Flows received its name because the mapping seeks to normalise a latent density function. As such, the authors could consider calling their method something like "RiemannianFlows" or "GeodesicFlows".

Minor fixes:
- Line 20: there shouldn't be a comma before "and"
  - Consider "these movements makes hand coding an infeasible approach. Therefore, learning techniques..."
- Line 29: should "model parameters optimization" be "model parameters' optimization"
- Line 129: the end of this sentence doesn't read well.
- Line 199: "a embedded" should be "an embedded"
- Line 244: should the n subscript in (x_{n,t}, \dot{x}_{n,t}) be m?
- Line 256: should "vectors point inwards the sphere are" be "vectors point inwards towards the sphere's center, are"
- Sub-captions would make figure 2 easier to interpret
- Figure 4 is referenced before Figure 3?
- Figure 4 is hard to read.
  + Consider making figure 4 only two figures wide with one figure

**Quality Of The Limitations Section:**

Limitations are addressed clearly

**Reviewer Expertise:**

4: The reviewer is confident but not absolutely certain that the evaluation is correct

**Robotics Focus:**

Sufficient demonstration on hardware

**Strengths And Weaknesses:**

**Strengths**
1. Mapping the system to a latent Riemannian manifold and encoding stability via a geodesic vector field is an elegant idea.
1. Their approach to calculating the differential of the inverse diffeomorphism by reversing the integration time interval is neat.
2. The real-world experiments verify that the method works in practice. I found the video in the supplementary material helpful for understanding the real-world experiments.
   - Perhaps the authors could add a link to the video in the experiments section?

**Weaknesses**

1. The experiments have not verified whether or not the latent space needs to be a Riemannian manifold to ensure that the stable dynamical system temporally satisfies geometric constraints. Can the diffeomorphism not learn a representation that preserves these temporal geometric constraints?
   - This paper assumes that the latent space should be a Riemannian manifold but more experiments are needed to support this claim. As a minimum, I would like to see RSDS compared to the original Euclideanizing Flows in Neumann et al., 2015. This is because the papers *modified* Euclideanizing Flows implementation maps the system to a latent Riemannian manifold (instead of Euclidean) and parameterises the diffeomorphism with a Neural ODE instead of a single-layer neural network with the layer resembling an approximated kernel machine.
   - As such, it is not possible to infer why the *modified* EuclideanizingFlows implementation performed poorly.
   - Comparing to results for EuclideanizingFlows with a diffeomorphism parameterised as a Neural Manifold ODE would allow direct comparison.
2. The paper restricts itself to diffeomorphisms parameterised as Neural Manifold ODEs.
   - Although it is perfectly fine to limit the scope of their work, I think that the authors could motivate the Neural Manifold ODEs better as it was not immediately clear to me why a Neural MODE was being used to learn the diffeomorphism.
   - My current understanding (after several reads) is that the Neural Manifold ODE is useful for calculating the pullback operator because the inverse of the diffeomorphism can be calculated by reversing the time integral. Although the authors clearly state the problem of computing the inverse due to geometric constraints at the end of the first paragraph in Section 3.3, they do not state that they use Neural Manifold ODEs to do this. In my opinion, the authors should motivate the use of Neural Manifold ODEs and add some more sign posting. For example, by say something like this at the start of Section 3.2:
     - "Given Eq. 5 and a dataset of $N$ human demonstrations, the goal of learning stable dynamics is now reduced to learning $\phi_{\theta}(x)$ and $\hat{k}\_{\gamma}(x)$. However, due to the geometric constraints arising from $\mathcal{M}$, calculating the inverse diffeomorphism $D_{y}\phi_{\theta}(y)$ is problematic. For this reason, we leverage Neural Manifold ODEs to learn the diffeomorphism because we can calculate the inverse diffeomorphism by simply reversing the integration time interval, as we shall see in Section XX."
3. Although not essential, the authors do not compare to experiments using the straightforward solution of calculating the pullback operator using the derivatives of the diffeomorphism.
   + Does this method provide greater accuracy?

**References**
Muhammad Asif Rana, Anqi Li, Dieter Fox, Byron Boots, Fabio Ramos, Nathan Ratliff Proceedings of the 2nd Conference on Learning for Dynamics and Control, PMLR 120:630-639, 2020.


**Summary Of Recommendation:**

Overall, this paper does a nice job combining various methods leading to a novel approach for learning motion dynamics from demonstrations. They provide theoretical results in the appendix and they test their method in two experimental settings. They test their method's ability at reproducing trajectories on the LASA dataset projected onto $\mathcal{S}^2$. That is, they test their method's ability to learn vector fields that encode letters projected onto a sphere. They then test its ability at reproducing trajectories from real demonstrations on a grasping task and a drawing task. The dynamic motion patterns required synchronised position-orientation trajectories and their results demonstrate that their method is capable of encoding this geometric structure. Although this paper extends the method from Euclideanizing Flows (Neumann et al., 2015), it presents a novel method for manually-designing stable dynamics in latent Riemannian spaces using geodesic vector fields and also provides a novel method for calculating the pullback operator by reversing the Neural Manifold ODE's integration time interval. The technical quality of this paper is also high. The authors detail key equations and intuitions in the paper and provide more rigorous theoretical contributions in the appendix.

---

> ### Author Response · Authors · 2022-08-24
> **Response to Reviewer AVAQ (1)**
>
> Thank you very much for your time reviewing our work and the thorough feedback!
> We are delighted to read that our approach on "mapping the system to a latent Riemannian manifold and encoding stability via a geodesic vector field is an elegant idea", and also that the reviewer found our solution to compute the pullback operator "neat"!  Below we address the issues raised in the review.
>
> **The experiments have not verified whether or not the latent space needs to be a Riemannian manifold**
>
> This is a very relevant point when learning diffeomorphisms. We addressed this question in the general response to the meta-reviewer summary, and we kindly refer the reviewer to it.
>
>
> **The paper restricts itself to diffeomorphisms parameterised as Neural Manifold ODEs**
>
> We chose to represent the learned diffeomorphism via Neural Manifold ODEs because of their generic formulation. State-of-the-art normalizing flows (NFs), such as the one used in EuclideanizingFlows, cannot be directly used as their layers are not designed to work on data with geometric constraints. Consider a diffeomorphism on the Sphere, its domain and co-domain are restricted on the spherical surface. Additionally, this function should be a one-to-one mapping and bidirectionally differentiable. These requirements are not fulfilled by using state-of-the-art NFs on Riemannian manifolds. However, Neural Manifold ODEs provide a general and technically-sound method to construct such a diffeomorphism on Riemannian manifolds by solving IVPs, although there may be computational limitations with this kind of constructions. We kindly refer the reviewer to our general response to the meta-reviewer's summary for further details on our choice of leveraging neural MODEs.
>
> On a related note, we added a text similar to the one suggested by the reviewer at the beginning of Section 3.2. to explicitly state that we use a Neural MODE to compute the pullback operator by reversing the time interval. The new text reads as follows:
>
> *Given the final RSDS learning framework in Eq.(5) and a dataset of $M$ demonstrations, the goal of learning stable dynamics on a Riemannian manifold reduces to learning $\psi_{\mathbf{\theta}}$, computing its pullback operator $D_{\mathbf{y}}\psi_{\mathbf{\theta}}^\star$, and subsequently estimating $\hat{k_{\mathbf{\gamma}}}(\mathbf{x})$.
> However, due to the geometric constraints arising from $\mathcal{M}$, learning a diffeomorphism and calculating the corresponding pullback operator are non-trivial problems.
> To address these challenges, we first leverage Neural MODEs [23] to build the diffeomorphism $\psi_{\mathbf{\theta}}$.
> In contrast to [23], we also propose a novel approach to compute the pullback operator that builds on reversing the time interval of the ODE integration, as we shall see in Section 3.3, avoiding to explicitly compute the corresponding inverse.
> Additionally, we propose a method to design Lyapunov-stable geodesic vector fields on a Riemannian manifold, which are leveraged to provide stability guarantees on the learned dynamical system, as explained in Section 3.1.*
>
> **Comparison against the straightforward solution of calculating the pullback operator using the derivatives of the diffeomorphism**
>
> As stated in Section 3.3, due to the geometric constraints, the pullback operator cannot be simply obtained by computing the inverse of derivatives of the diffeomorphism. Such computation involves solving a constrained optimization problem, which requires an iterative process in most Riemannian manifolds, where manifold-specific constraints are nonlinear. Note that the straightforward solution provided in Eq. (7) is only specific for the Sphere manifold $\mathcal{S}^n$, since the constraints can be formulated as linear equations. For this reason, we apply our more general solution via the adjoint method for all our experiments. But both methods provide the same solution for the pullback operator when considering $\mathcal{S}^n$.

---

> > ### Author Response · Authors · 2022-08-24
> > **Response to Reviewer AVAQ (2)**
> >
> > **Incorrect statement about Euclideanizing Flows in Neumann et al., 2015**
> >
> > We believe that the reviewer may be referring to the work proposed by Rana et al, 2020 (i.e. Euclideanizing Flows), instead of Neumann et al, 2015. Please correct us if we misread your comment.
> > Despite this, we agree with the reviewer's comment, Euclideanizing Flows generally considers that the dynamical system evolves on a Riemannian manifold and the associated stable system is designed on an Euclidean space. We introduced this clarification in the revised version of the paper.
> > However, we would like to point out that EuclideanizingFlows assume that there exists a function that maps any point of the Riemannian manifold to an (ambient) Euclidean space, and therefore the diffeomorphism is still learned between two Euclidean spaces. This assumption seems to be crucial in EuclideanizingFlows, as the class of diffeomorphisms considered in the paper applies geometry-unaware operations on the input layer. Such operations can certainly be applied to Euclidean data, but they cannot be employed directly on the input layer when the system state lies on a Riemannian manifold as they disregard the geometric constraints arising from $\mathcal{M}$.
> >
> > In other words, the parametrization of the input layer of the diffeomorphism proposed in EuclideanizingFlows is a sort of approximated kernel machine (as pointed out by the reviewer), where the kernel feature maps assume Euclidean inputs (or they disregard the input geometry). Similarly, the inverse of the proposed diffeomorphism does not necessarily provide outputs that comply the geometric constraints of Riemannian data.
> > Last but not least, the experiments shown in the EuclideanizingFlows paper did not report tests with data lying on Riemannian manifolds.
> > Consequently, we think that EuclideanizingFlows do not fully address the problem of learning stable dynamical systems on Riemannian manifolds.
> > Furthermore, as stated by reviewer eXxr, EuclideanizingFlows assume that the Riemmanian manifolds are isomorphic to Euclidean space, which does not generalize for several Riemannian manifolds of interest (e.g., compact manifolds such as the Sphere are not diffeomorphic to Euclidean space).
> > We kindly refer the reviewer to our general response to the meta-reviewer's summary where we provide further theoretical insights on the importance of considering the Riemannian geometry when learning diffeomorphisms.
> >
> > **Statement about learning diffeomorphisms with normalising flows**
> >
> > We agree that our approach resembles the formulation of normalizing flows on Riemannian manifolds, but it is not strictly concerned about fitting a latent density function. Because of this, we rewrote some few parts of the paper to reflect this difference.

---

### Official Review · Reviewer_SPAS · 2022-07-30

**Originality:** Very Good
**Technical Quality:** Excellent
**Clarity Of Presentation:** Excellent
**Impact:** 4

**Recommendation:**

Weak Accept: I recommend accepting the paper, but will not argue for my recommendation if the majority of other reviewers have a different opinion.

**Summary:**

The paper presents a method to learn robot motions that are restricted to specific Riemannian manifolds using stable dynamical systems and diffeomorphisms. The proposed technique combines recent developments from neural dynamics and geometry-aware formulations of robot motions in order to ensure both stability as well as geometric constraints on trajectories (e.g., trajectories belong to a specific Riemannian manifold). Experiments demonstrate that proposed method is better at respecting geometric constraints of the trajectories, compared to a recent method capable of learning stable dynamical systems via diffeomorphisms.

**Issues:**

Please go over my comments from above and provide clarifications where appropriate, and push back on points that reflect my misunderstanding.

**Quality Of The Limitations Section:**

Limitations are addressed clearly

**Reviewer Expertise:**

4: The reviewer is confident but not absolutely certain that the evaluation is correct

**Robotics Focus:**

Sufficient demonstration on hardware

**Strengths And Weaknesses:**

### Strengths

- The proposed methodology is well explained and all necessary details to understand the core ideas are provided. Further, I appreciated the additional comprehensive background and details in the complementary material.
- The related work is section does a pretty good job of discussing some of the most relevant methods for dynamical-systems based LfD, and positions the paper well within this subfield.
- The proposed method is theoretically sound and brings together a collection of recent ideas from different related communities in machine learning, dynamical systems, and robotics.
- The experiments conclusively demonstrate that proposed method can learn stable dynamical systems on a Riemannian manifold and that the reproductions obey the inherent geometric constraint.

### Weakness

- While the paper does a fantastic job and detailing the theoretical differences between the proposed approach and that of existing approaches, I felt that the expressed motivation for robotics was somewhat unsatisfactory. Indeed, examples of manifolds that are relevant to robotics are suggested (e.g., $SO(3)$ and $SE(3)$), the practical benefits of the claimed theoretical properties remains elusive.

- On a related note: While benefits over existing methods that do not obey geometric constrains is clear (and demonstrated), benefits over methods that rely on Lie groups (e.g., Ref [10]) is somewhat unclear. Indeed, it is stated that not all manifolds are easily expressed as Lie groups. But this argument demonstrated in experiments. In fact, only one of the related approach was used as a baseline in the experiments (that too only in the simulated dataset).

- The robot experiment involved relatively simplistic tasks that are unlikely to demonstrate the benefits of the nuanced theoretical differences between the proposed methods and the existing methods.  This problem is exacerbated by the fact that the physical robot experiments did not involve any comparisons with existing approaches and merely serves as a proof-of-concept of the approach. I find it hand to imagine how existing methods will fail in these tasks.

- It might be helpful to slightly expand the discussion of related work to include a class of methods known as Riemannian Motion Primitives (RMPs) (e.g., [1]]) and Geometric Fabrics [3]. I do not believe that these methods subsume the contributions of the proposed method but these methods employ very similar structures in the policy and it would be worthwhile to understand how the proposed approach differs from them.

- The technical advances over Ref [21] (the Manifold ODE paper) could be made more explicit so that the reader will understand why one can not view the proposed method as a trivial application of the technique in Ref. [21] to robotics tasks.

### References

1.  Rana, M. A., Li, A., Ravichandar, H., Mukadam, M., Chernova, S., Fox, D., ... & Ratliff, N. (2020, May). Learning reactive motion policies in multiple task spaces from human demonstrations. In Conference on Robot Learning (pp. 1457-1468). PMLR.
2. Van Wyk, K., Xie, M., Li, A., Rana, M. A., Babich, B., Peele, B., ... & Ratliff, N. D. (2022). Geometric fabrics: Generalizing classical mechanics to capture the physics of behavior. IEEE Robotics and Automation Letters, 7(2), 3202-3209.


**Summary Of Recommendation:**

The paper is well written and provides an elegant theoretical approach to learn stable robot motions in complex spaces. The theoretical presentation and exposition is commendable and the experiments serve to fortify some of the claims made in the paper. My only criticism stems from the fact that the practical benefits of the nuanced theoretical are either i) mentioned but not substantiated in the experiments, or ii) not mentioned at all.

---

> ### Author Response · Authors · 2022-08-24
> **Response to Reviewer SPAS (1)**
>
> We are very grateful for your time reviewing our work and the provided suggestions!
> We appreciate the positive feedback about our paper, and are glad to read that our approach "is theoretically sound" and that our experiments "fortify some of the claims in the paper"!  Below we address the issues raised in the review.
>
> **Motivation for robotics and practical benefits**
>
> To better motivate our RSDS approach from a practical point of view, we added a couple of simulated robotic experiments where both RSDS and the baseline are compared. It is clearly shown that our Riemannian formulation provides stronger stability guarantees than the baseline when learning relatively-simple orientation patterns on $\mathcal{S}^3$. We believe that these results support the motivation of properly considering the geometry of the data when learning stable dynamical systems. We kindly refer the reviewer to our general response where we provide the details of the additional experiments and new figures in the attached PDF file.
>
>
> **Benefits over Lie-groups-based methods**
>
> We agree with the reviewer about the fact that our experiments were focused on learning vector fields of the robot end-effector motion in $\mathbb{R}^3 \times \mathcal{S}^3$, which could equivalently be formulated as learning vector fields on $\operatorname{SE}(3)$, endowed with a Lie group structure, as proposed in [10]. Our experiments were designed to show that our approach can be applied to this well-known problem of robot dynamic motion learning.
> However, we would like to emphasize that assuming a Lie-group structure may limit the range of applications of [10], as it imposes some constraints on the problem domain. We hypothesize that our framework may be leveraged in problems concerned about learning vector fields on the manifold of symmetric positive matrices $\mathcal{S}_{++}$. For example, recent works on robot learning [a] leveraged such a manifold to learn posture-dependent skills via manipulability ellipsoids, showing a possible extension of our work. Besides robot motion generation, we believe that robot perception may also benefit from our more general approach. For instance, human and/or objects motion tracking is a key aspect when designing autonomous robots that dynamically interact with the environment. Therefore, learning dynamical models for motion tracking (see [b]) may be another potential application of our general Riemannian framework. Thus, we believe that the proposed general Riemannian formulation opens the door to a broader range of potential applications beyond robot end-effector motion generation, unlike [10].
>
> [a] N. Jaquier, et al. Geometry-aware manipulability learning, tracking, and transfer, IJRR, 2021.
>
> [b] G. Cheng and B. C. Vemuri. A novel dynamic system in the space of SPD matrices with applications to appearance tracking. SIAM Journal on Imaging Sciences, 2013.
>
>
> **Robotic experiments**
>
> As pointed out above, we carried our two additional simulated robotic experiments to show the practical benefits of our approach in terms of stability guarantees. We kindly refer the reviewer to the general response and rebuttal attachment for further details.

---

> > ### Author Response · Authors · 2022-08-24
> > **Response to Reviewer SPAS (2)**
> >
> > **Discussion about RMPs and Geometric Fabrics**
> >
> > Our RSDS approach may be seen as learning a control policy represented by a first-order dynamical system. As such, RSDS may be employed as a motion policy into the RMPs framework, as RMPs provide a geometric robot motion structure that can combine several motion polices with associated Riemannian metrics. In other words, we may fuse RSDS skills with additional motion policies, like obstacle avoidance behaviors, under the RMPs framework, which has also been exploited in [c].
> > Geometric Fabrics are a more general and powerful approach than RMPs as they build on Finsler geometries, which can be seen as a generalization of Riemannian geometry for dynamical systems.
> > On a related note, RSDS represents a learned first-order dynamic motion policy that does not explicitly consider a specific type of underlying mechanical system. In contrast, RMPs build on Riemannian geometric control to design robot motion policies whose geometry is characterized by, e.g., the associated inertia matrix of the robot dynamics.
> > We added the above discussion to the paper in Section 1, as a new paragraph starting from line 64. The text reads as follows:
> >
> > *Note that Riemannian geometry has been also leveraged to design robot motion policies that build on the geometry of classical mechanical systems, as proposed in recent works on Riemannian Motion Policies [25], and more recently in a generalization called Geometric Fabrics [26]. In this context, RSDS may be seen as learning a control policy represented by a first-order dynamical system, which may be employed as a motion policy into the RMPs framework, as RMPs provide a geometric robot motion structure that can combine several motion polices with associated Riemannian metrics. In other words, we may fuse RSDS skills with additional motion policies, like obstacle avoidance behaviors, under the RMPs framework, similar to [27]. Note that RSDS represents a learned first-order dynamic motion policy that does not explicitly consider a specific type of underlying mechanical system. In contrast, RMPs build on Riemannian geometric control to design robot motion policies whose geometry is characterized by, e.g., the associated inertia matrix of the robot dynamics.*
> >
> > [c] M. A. Rana, A. Li, D. Fox, S. Chernova, B. Boots, and N. Ratliff. Towards coordinated robot motions: End-to-end learning of motion policies on transform trees, IROS, 2021.
> >
> >
> > **Technical advances over Neural MODE**
> >
> > When comparing our work to Neural MODEs, the technical advances are: (1) computation of the pullback operator (Sec. 3.3), and (2) design of Lyapunov-stable geodesic vector fields (Sec. 3.1). To better reflect this in our paper, we modified the opening paragraph of Section 3.2 as follows:
> >
> > *Given the final RSDS learning framework in Eq.(5) and a dataset of $M$ demonstrations, the goal of learning stable dynamics on a Riemannian manifold reduces to learning $\psi_{\mathbf{\theta}}$, computing its pullback operator $D_{\mathbf{y}}\psi_{\mathbf{\theta}}^{\star}$, and subsequently estimating $\hat{k_{\mathbf{\gamma}}}(\mathbf{x})$.
> > However, due to the geometric constraints arising from $\mathcal{M}$, learning a diffeomorphism and calculating the corresponding pullback operator are non-trivial problems.
> > To address these challenges, we first leverage Neural MODEs [23] to build the diffeomorphism $\psi_{\mathbf{\theta}}$.
> > In contrast to [23], we also propose a novel approach to compute the pullback operator that builds on reversing the time interval of the ODE integration, as we shall see in Section 3.3, avoiding to explicitly compute the corresponding inverse.
> > Additionally, we propose a method to design Lyapunov-stable geodesic vector fields on a Riemannian manifold, which are leveraged to provide stability guarantees on the learned dynamical system, as explained in Section 3.1.*

---

### Official Review · Reviewer_hQF2 · 2022-07-31

**Originality:** Very Good
**Technical Quality:** Fair
**Clarity Of Presentation:** Very Good
**Impact:** 3

**Recommendation:**

Weak Reject: I recommend rejecting the paper, but will not argue for my recommendation if the majority of other reviewers have a different opinion.

**Summary:**

The paper considers the problem of learning stable dynamical systems on manifolds. The main approach proposed in the paper combines existing methods in 1) learning stable dynamical system via diffeomorphisms and 2) normalizing flows on manifolds. The paper additionally proposes solution for computing the pullback operator. The proposed method is validated on a simulated task and 2 real robot tasks.

**Issues:**

(The list below is the same as **Weaknesses**)
- **Global asymptotic stability.** The proposed method is claimed to "ensure global asymptotic stability", which is the main motivation behind the diffeomorphism parameterization. It seems, however, that *global* asymptotic stability, at least for a single equilibrium point, may not be possible. This is due to the Poincaré–Hopf theorem: the sum of the Poincaré indices over all the isolated zeroes of a vector field must be equal to the Euler characteristic of the manifold. Globally stable local equilibrium point means a single zero with index 1. However, for, e.g., 2-sphere which has Euler characteristic 2, there must exist at least another zero. This means that globally asymptotically stable is not achievable for such a manifold. I might be wrong here so maybe the authors can elaborate more on the existence of globally asymptotic equilibrium points.
- **Usage of CNF.** In the conclusion, the authors mentioned computational limitation caused by the Neural MODE. Can the authors provide more explanation on why they still choose this diffeomorphism parameterization rather than others in the normalizing flow literature? Are other parameterizations infeasible for learning diffeomorphisms on manifolds?
- **Comparison with baseline.** It would be nice if the author can also compare their proposed method to the baseline in the real robot experiment.

**Quality Of The Limitations Section:**

Limitations are addressed clearly

**Reviewer Expertise:**

3: The reviewer is fairly confident that the evaluation is correct

**Robotics Focus:**

Sufficient demonstration on hardware

**Strengths And Weaknesses:**

### Strengths
- **Clarity.** Despite being technically involved, the paper is well-written. The authors provided background on differentiable geometry in a concise and easy-to-follow way. Readers with little background on differential geometry should be able to follow along.
- **Organization.** The paper is well-organized. The author provided concise and useful differentiable geometry background and normalizing flow. The technical section provides detailed description of the proposed method. The equations are well-explained.
- **Experiments.** The authors showed significant improvement over the baseline algorithm on a simulated task. The paper also contains a real robot experiment, showing that the proposed method is scalable to real hardware.
### Weaknesses
- **Global asymptotic stability.** The proposed method is claimed to "ensure global asymptotic stability", which is the main motivation behind the diffeomorphism parameterization. It seems, however, that *global* asymptotic stability, at least for a single equilibrium point, may not be possible. This is due to the Poincaré–Hopf theorem: the sum of the Poincaré indices over all the isolated zeroes of a vector field must be equal to the Euler characteristic of the manifold. Globally stable local equilibrium point means a single zero with index 1. However, for, e.g., 2-sphere which has Euler characteristic 2, there must exist at least another zero. This means that globally asymptotically stable is not achievable for such a manifold. I might be wrong here so maybe the authors can elaborate more on the existence of globally asymptotic equilibrium points.
- **Usage of CNF.** In the conclusion, the authors mentioned computational limitation caused by the Neural MODE. Can the authors provide more explanation on why they still choose this diffeomorphism parameterization rather than others in the normalizing flow literature? Are other parameterizations infeasible for learning diffeomorphisms on manifolds?
- **Comparison with baseline.** It would be nice if the author can also compare their proposed method to the baseline in the real robot experiment.

**Summary Of Recommendation:**

The paper considers the problem of learning stable dynamical systems on manifolds. The main approach proposed in the paper combines existing methods in 1) learning stable dynamical system via diffeomorphisms and 2) normalizing flows on manifolds. The paper additionally proposes solution for computing the pullback operator.

The paper is well-written. The authors provided background on differentiable geometry in a concise and easy-to-follow way. Readers with little background on differential geometry should be able to follow along. The authors showed significant improvement over the baseline algorithm on a simulated task. The paper also contains a real robot experiment, showing that the proposed method is scalable to real hardware.

However, there is some concern on the theoretical soundness of the paper. I am happy to increase my score if my concern is addressed.

---

> ### Author Response · Authors · 2022-08-24
> **Response to Reviewer hQF2**
>
> Thank you very much for your review! We are very pleased to read that the reviewer found our paper well-written, explained and organized! Below we address the issues raised in the review.
>
> **Global asymptotic stability**
>
> Thank you for raising this important point! We agree with the reviewer's observation regarding the global asymptotic stability analysis. We kindly refer the reviewer to our general response above where we discuss our assumptions and changes made in the paper.
>
>
> **Usage of Neural MODE**
>
> We provide an elaborated answer to this point in the general response to the meta-reviewer summary, we kindly refer the reviewer to it.
>
>
> **Comparison with baseline**
>
> As suggested by the reviewer, we added a couple of simulated robotic experiments where we compare the performance of our RSDS approach against the baseline. The reported experiments show that our approach provide stronger and more technically-sound stability guarantees w.r.t the baseline, which diverges when learning vector fields on $\mathcal{S}^3$. We kindly refer the reviewer to the general response where we describe in detail the additional experiments and provide some figures in the rebuttal attachment.

---

> > ### Comment · Reviewer_hQF2 · 2022-08-27
> > **Thank you for your response**
> >
> > Thank you for your response! I appreciate the new discussion on geodesic vector field and cut locus. However, I still find the "quasi-global asymptotic stability" statement a little unsatisfying.
> >
> > With that, I will maintain my score. However, I am fine with accepting this paper if the majority of the reviewers vote so, since I agree that this paper has some good merits and potential impact to robotics.

---

### Official Review · Reviewer_eXxr · 2022-08-05

**Originality:** Fair
**Technical Quality:** Good
**Clarity Of Presentation:** Good
**Impact:** 3

**Recommendation:**

Weak Accept: I recommend accepting the paper, but will not argue for my recommendation if the majority of other reviewers have a different opinion.

**Summary:**

This paper builds on previous methods of learning stable dynamical systems for imitation learning, by starting with a stable vector field, and morphing it according to demonstrations, while remaining stable. Previous methods, such as Euclideanizing flows, rely on diffeomorphisms map to isomorphisms of Euclidean space. This paper defines invertible functions which operate on a wider class of Riemannian manifolds by using the differentiable integrator from Manifold Neural ODEs (MNODEs), to enforce the vector field on a specified manifold. The paper empirically evaluates the usage of the diffeomorphism created by the MNODE on the LASA dataset on S^2 and two real robot environments.

**Issues:**

-> run experiments on the whole LASA dataset, not just 4 cherry-picked characters;
-> report on runtime;
-> further clarify (in writing) motivations where the manifold variant is useful -- the LASA on S^2 is slightly contrived;
-> include Zhi et al. and Urain et al., which are very relevant.

**Quality Of The Limitations Section:**

Limitations are addressed clearly

**Reviewer Expertise:**

5: The reviewer is absolutely certain that the evaluation is correct and very familiar with the relevant literature

**Robotics Focus:**

Sufficient demonstration on hardware

**Strengths And Weaknesses:**

The idea is sound, and the execution is good. Methods to use diffeomorphisms to maintain stability has been estabilished in (Rana et al, L4DC, 2020; Urain et al., IROS, 2020; Zhi et al, L4DC, 2022), and the proposed extension via Manifold Neural ODEs (MNODEs) is quite logical. The execution is also good, with real robot experiments.

I'm less enthusiastic with the motivation here. I've dabbled with Manifold Neural ODEs, and they are much slower than differentiable integrators from the torchdiffeq library. I'm not sure when there would be a setup where the fairly minor performance gains are worth the additional time expenditure. Regardless, I think it is useful to extend diffeomorphism-based imitation learning to restricting it to a wider range of Riemannian manifolds, and I vote for acceptance.


**Summary Of Recommendation:**

The method is sound, and the experiments are the correct ones. More experiments and results on the LASA dataset could be included, the "S" is a classic, and there are quite a lot of characters -- why only select such a small number of letters? The results presented are graphical, it would be good to see some numbers over more characters.

The limitation sections make note that NMODEs are slow. Runtime is quite important here -- if it is sufficiently fast, perturbance can be accounted for online. Please quantify and report on it.

Additionally, Zhi et al. "Diffeomorphic Transforms for Generalised Imitation Learning", L4DC 2022 and Urain et al., "ImitationFlow: Learning Deep Stable Stochastic Dynamic Systems by Normalizing Flows" IROS 2020 are relevant references that should be included.

---

> ### Author Response · Authors · 2022-08-24
> **Response to Reviewer eXxr**
>
> We would like to thank you for taking the time to review our work! We are delighted to read that "our idea is sound and the execution is good"! Below, we address some of the key concerns raised as part of the review.
>
> **Further experiments on the LASA dataset**
>
> As suggested by the reviewer, we carried out three additional experiments on new letters from the LASA dataset, specifically: $\mathsf{S}$, $\mathsf{SharpC}$ and $\mathsf{Spoon}$, whose trajectories significantly differ from the datasets used in our paper. The corresponding results are shown in *Figs. 1-4* of the attached PDF file and also added to *Appendix F.1*. Note that these additional experiments further support the results provided in the paper. Figure 1 clearly shows that all blue and black rollouts of RSDS closely match the demonstrations and converge to the attractor, unlike the baselines. Figure 2 shows that RSDS provides the highest success rate and lower (or equal) MSE with respect to the baselines. Figure 3 shows that a large number of the Euclideanflows trajectories failed to converge despite the projection, however all the RSDS trajectories succeeded for the additional datasets as well. Note that the unstable behavior of the Euclidean approach may be attributed to spurious attractors arising in the learned vector fields, as shown in Fig. 4. These points often appear when working with geometry-unaware solutions.
>
> **Motivation of Riemannian manifold extension and Neural MODEs**
>
> As pointed out by the reviewer, one of the limitations of our approach is its slow runtime (as discussed in the limitations section of our paper). We believe that this problem may be resolved in the future when faster integrators on Riemannian manifolds are developed.
> As mentioned in our general response, a more immediate solution involves reducing the number of coordinate charts used in Neural MODEs.
> However, the importance of having a technically-sound Riemannian approach can also be verified in robotic settings. For this, we would like to refer the reviewer to the additional simulated robotic experiments (see Figs. 5 and 6 in the attached file), which clearly show that a formal Riemannian framework is crucial for stability guarantees, even for fairly simple orientation patterns. The unstable behaviors can be attributed to the fact that geometry-unaware approaches employ a Euclidean latent space on which the diffeomorphism is learned. As explained in our general answer to the meta-reviewer, this approach is flawed as diffeomorphisms preserve the topological structure of the manifold. Therefore, it is not possible to construct a diffeomorphism between compact manifolds and Euclidean space. As such, stability guarantees cannot be accomplished. In contrast, our RSDS approach is able to learn stable dynamical systems, as shown in several LASA datasets on $\mathcal{S}^2$ and in the simulated and real robotic tasks reported in the attached file and the original paper.
>
> **Runtime of our approach**
>
> We added the runtime of RSDS and EuclideanizingFlows for three different experiments in the attached file (see Table 1). The same table was added to Appendix F.1.
>
> **Additional references**
>
> We thank the reviewer for referring us to the works of Zhi et al, and Urain et al. We added them to the related works that exploit diffeomorphisms in the context of imitation learning of robot motions in Euclidean spaces, reviewed in Section 1 (lines 33 and 40).

---

### Meta-Review · Area_Chair_9S1c · 2022-08-13

**Recommendation:** Accept (Poster)
**Confidence:** 4

**Metareview:**

The reviewers commented positively about the theoretical derivation of the method, its main properties and the experiments performed. However, there are also some important issues that need to be addressed in the rebuttal phase:

1. Discuss the motivation of Neural Manifold ODEs given that they are slower than other integrators
2. Discuss the global asymptotic stability for a single equilibrium point
3. Discuss additional references as pointed by reviewers
4. Better motivate the practical benefits of the method to robotics applications
5. Add comparison to Euclideanizing flows with a NM-ODE to justify the assumption of a latent Riemannian manifold

============================
Post rebuttal update

The authors have addressed most of the concerns with the updated version of the paper. Reviewers had mixed opinions about the paper but still consider the formulation interesting and useful. For this reason I recommend accepting the paper.

**Best Paper Nomination:**

No

---

> ### Author Response · Authors · 2022-08-24
> **General response to all reviewers and meta-reviewer (1)**
>
> We would like to thank the reviewers and meta-reviewer for their highly useful recommendations on the previous version of the manuscript. We have addressed the reviewers' comments and improved our paper accordingly. We organize our response as follows: *(1)* we provide a general reply that addresses the five points highlighted by the meta-reviewer (some of which are common across reviewers), and *(2)* we provide reviewer-specific replies. For the sake of simplicity, **we also attach a PDF file that includes all the new experiments carried out as suggested in the the reviewers' feedback.**
>
> **Motivation of Neural Manifold ODEs**
>
> We agree that our motivation for using Neural MODEs did not come fully clear through the text. Note that designing and learning a diffeomorphism between Riemannian manifolds is a non-trivial problem. One of the first attempts to address this problem considered projecting the data onto Euclidean space and then employed state-of-the-art normalizing flows (e.g. a Real NVP), whose solution was projected back to the manifold [a]. Such an approach is flawed as this strictly requires Riemannian manifolds to be diffeomorphic to Euclidean space (compact manifolds such as Tori and the Sphere are not). One could then think of designing manifold-specific normalizing flows such that the corresponding neural network is a diffeomorphism, which was proposed in [b]. However, such manifold-specific solutions do not generalize to different types of manifold topologies. Last but not least, the networks employed in state-the-art methods such as EuclideanizingFlows are parametrized by geometry-unaware layers, which assume the training data are Euclidean, therefore their use in Riemannian settings is flawed.
>
> Because of the above reasons, we chose neural MODEs to parametrize our diffeomorphism. Neural MODEs are more general as they build on solutions of ODEs evolving on a Riemannian manifold. As mentioned in the limitation sections of our paper, such general approach comes with a price: the computational cost of solving an ODE.  We believe that this problem may be resolved in the future when faster integrators on Riemannian manifolds are developed (which is currently a topic of interest). A more immediate solution involves reducing the number of coordinate charts used in Neural MODEs, which significantly reduces our approach runtime.
>
>
>
> We added the following text in Section 1 (starting from lines $60$) to more clearly motivate our choice. The new text reads as follows:
>
> *Unlike works assuming Riemannian manifolds that are diffeomorphic to the Euclidean space [25], or manifold-specific diffeomorphisms built on specialized neural networks [26], our RSDS approach leverages a more general formulation to construct diffeomorphisms based on solutions of ODEs evolving on arbitrary Riemannian manifolds [23].*
>
> [a] M. Gemici, D. J. Rezende, and S. Mohamed. Normalizing flows on Riemannian manifolds. 2016.
>
> [b] D. J. Rezende, G. Papamakarios, S. Racani`ere, M. S. Albergo, G. Kan- war, P. E. Shanahan, and K. Cranmer. Normalizing flows on tori and spheres. ICML, 2020.

---

> > ### Author Response · Authors · 2022-08-24
> > **General response to all reviewers and meta-reviewer (2)**
> >
> > **Global Asymptotic Stability**
> >
> > We agree with the observation about the relationship between the Euler characteristic of a manifold $\mathcal{M}$ and the global convergence to a single attractor on $\mathcal{M}$. As the reviewer hQF2 pointed out, the Poincaré-Hopf theorem establishes that for any vector field $v$ on $\mathcal{M}$, the sum of the Poincaré indices over all the isolated zeroes is equal to the Euler characteristic $\chi(\mathcal{M})$. Because of this, we agree that it is not possible to guarantee *global* asymptotic stability on the Sphere.
> >
> > However, we would like to elaborate our assumption behind the design of the geodesic vector field. First, let us consider the case of the Sphere manifold $\mathcal{S}^n$. For a geodesic vector field defined on $\mathcal{S}^n$, the remaining isolated zero corresponds to the cut locus of the attractor point $\operatorname{Cut}(\mathbf{x}^{\ast})$ where the vector field converges to. Our assumption is that a geodesic vector field converges to a single attractor $\mathbf{x}^{\ast}$ for all points on $\mathcal{M}$ except the attractor's cut locus $\operatorname{Cut}(\mathbf{x}^{\ast})$. Such an assumption implies that theoretically we may only guarantee "*quasi-global* asymptotic stability". Although our assumption might be restrictive when designing geodesic vector fields on compact Riemannian manifolds (e.g., the Sphere), there exist several non-compact manifolds without cut locus (e.g., the manifold of symmetric positive definite matrices) [c], for which the aforementioned assumption is not necessary.
> >
> > In practice, we can assign non-zero tangent vectors at the cut locus, which allow us to provide *global* asymptotic stability on the latent manifold. And the asymptotic stability of the target dynamical system is then inherited from the latent dynamical system through the learned diffeomorphism.
> >
> > We added the above explanation in App. B, and relaxed our claim about global asymptotic stability in the main paper, by stating that for certain manifolds our RSDS approach can only guarantee *quasi-global* asymptotic stability guarantees due to the Poincaré-Hopf theorem.
> >
> > [c] X. Pennec. Manifold-valued image processing with SPD matrices. 2020.
> >
> >
> > **Additional References**
> >
> > We added to our related work the papers [d,e] as suggested by reviewer eXxr and a short discussion about the similarities and differences of our work with respect to RMPs [f] and Geometric Fabrics [g], as suggested by reviewer SPAS. Details about the new text are given in the responses to the corresponding reviewers.
> >
> > [d] Zhi et al. “Diffeomorphic Transforms for Generalised Imitation Learning”, L4DC 2022.
> >
> > [e] Urain et al., “ImitationFlow: Learning Deep Stable Stochastic Dynamic Systems by Normalizing Flows” IROS 2020.
> >
> > [f] Rana, M. A., et al. Learning reactive motion policies in multiple task spaces from human demonstrations. CoRL, 2020.
> >
> > [g] Van Wyk, K., et al. Geometric fabrics: Generalizing classical mechanics to capture the physics of behavior. RA-L, 2022.
> >
> >
> > **Practical Benefits of our Method**
> >
> > As mentioned above, when learning a diffeomorphism that provides stability guarantees, there exist some theoretical aspects that justify the choice of having a Riemannian latent space, such as preserving manifold compactness. When considering geometry-unaware solutions, such as those relying on Euclidean latent spaces or building on projection methods, stability guarantees are severely compromised. To experimentally show the importance of geometry-aware formulations like RSDS on robotic tasks, we designed two experiments where a simulated Franka-Emika robot is required to track a vector field on $\mathcal{S}^3$ learned from synthetic demonstrations. During reproduction, the robot end-effector starts from randomly sampled points on $\mathcal{S}^3$, and it then follows the learned vector field. As Figs. 5 and 6 of the attached PDF file show, our RSDS approach always converges to the target while the modified version of EuclideanizingFlows diverges in some cases. We believe that this confirms the practical benefits of our RSDS method in robotic settings, as geometry-unaware approaches fail to provide stability guarantees on Riemannian manifolds, and therefore they cannot be employed in practical settings without taking special care of unstable cases.

---

> > > ### Author Response · Authors · 2022-08-24
> > > **General response to all reviewers and meta-reviewer (3)**
> > >
> > > **Assumption of a Latent Riemannian Manifold and Corresponding Comparison**
> > >
> > > As pointed out above, when learning a diffeomorphism, one must consider that this mapping function preserves the topological structure of the manifold, e.g., its compactness and connectedness. This means that compact manifolds such as the Sphere or Tori cannot be diffeomorphic to the Euclidean space, and therefore learning a diffeomorphism between them is not possible. Therefore, in the context of learning stable dynamical systems on Riemannian manifolds via diffeomorphisms, it is necessary that the latent space be diffeomorphic to the original manifold, which in turn is required for providing stability guarantees. Because of this, we believe that a comparison against an approach that considers a “diffeomorphic mapping” between a Riemannian manifold and Euclidean space is not necessary as the approach is theoretically flawed.